# Monoclonal Antibodies in Metastatic Gastro-Esophageal Cancers: An Overview of the Latest Therapeutic Advances

**DOI:** 10.3390/ijms26031090

**Published:** 2025-01-27

**Authors:** Foteini Kalofonou, Melpomeni Kalofonou, Foteinos-Ioannis Dimitrakopoulos, Haralabos Kalofonos

**Affiliations:** 1Department of Medical Oncology, The Royal Marsden NHS Trust, London SW3 6JJ, UK; 2Centre of Bio-Inspired Technology, Department of Electrical and Electronic Engineering, Imperial College London, London SW7 2AZ, UK; 3Division of Oncology, Department of Medicine, Medical School, University of Patras, 26504 Rion, Greece; 4Division of Oncology, Olympion General Clinic, 26443 Patras, Greece

**Keywords:** monoclonal, antibodies, gastric, esophageal, cancer, hybridomas, immunoconjugates, biotechnology, progression free survival, overall survival, HER2, PDL-1, VEGF, trastuzumab, ipilimumab, nivolumab, zolbetuximab, margetuximab, pembrolizumab, tislelizumab, PD-1, CTLA4

## Abstract

Monoclonal antibodies (mAbs) have completely changed the face of oncology over the last 50 years, and they have contributed to a major breakthrough in terms of cancer therapy. Esophageal and gastric cancers, the eighth and fifth most commonly diagnosed types of cancer worldwide, respectively, have lately, been managed more effectively, with the introduction of new therapeutic treatment strategies, especially mAbs. Combination treatments and new molecules have changed the face of the disease, while more therapies are getting approved on a daily basis. This review aims to analyse the major up-to-date clinical trials using mAbs and immunotherapy for the treatment of advanced gastro-esophageal cancers.

## 1. Introduction

Since the initial introduction of the first mAb in 1975 and the first acquirement of its full license in 1986, the field of mAbs has exponentially developed and expanded. MAbs represent an innovative method of targeting specific mutations and protein structure malformations, in a wide range of diseases and conditions [1]. Due to the recent major advancements in molecular biology and especially in genetics, as well as in bio-technology, humanised mAbs are rapidly growing into pharmaceutical molecules with great clinical application and efficacy [2].

MAbs derive from a single B-lymphocyte clone, binding to the same antigen, first generated in mice, using a hybridoma technique [3]. These mAbs bind to a specific epitope, which is the antigenic determinant of the antigen. MAbs generation using the hybridoma technology can be performed through in vitro or in vivo process, with these hybridomas containing initially a mixture of antibodies, which derive from many different primary B-lymphocyte clones [4].

MAbs have evolved over the years, and the immunoconjugate technology used provides the attachement to a pharmaceutical component or drug, which can carry them around, via a chemical linker. These antibody–drug conjugates (ADCs) were designed to selectively deliver the cytotoxic chemical component of the drug directly to the target cells [5]. ADCs are often fragmented using separate analyses of individual analytes. As such, it is difficult to compare among ADCs, in terms of pharmacokinetic characteristics, which can be crucial, in order to better understand the way an ADC complex can navigate and target its lesion and its efficacy [6].

Gastric cancer (GC) is the fifth most common cancer worldwide, with more than one million new cases diagnosed annually, while esophageal cancer (EC) is the eighth most common cancer, with more than 600,000 new cases diagnosed annually. Both have been associated with lifestyle, environmental factors, diet or infections (*H. pylori*) [7]. The treatment of these clinical entities, especially in the metastatic setting, remains a very demanding challenge for every physician, and the need for new pharmaceutical agents has remained unmet for a very long time.

In more recent years, however, with the latest breakthroughs in molecular profiling and the better understanding of the tumour biology, mAbs have been broadly incorporated into the gastro-esophageal cancer treatment. Advances in sequencing have revealed mutations and aberrations in a panel of targetable genes, including *EGFR*, which was found to be overexpressed in 60–86% of GC or gastro-esophageal tumours and in 50–70% of EC [8]. HER2 overexpression in GC (7–34%), which appeared to be associated with poorer prognosis, is now linked to improved outcomes among these patients, thanks to the use of trastuzumab, a mAb directed against the extracellular HER2 domain [9]. Angiogenesis is a predominant element in tumour development and evolution and is normally absent in healthy tissues, except for particular physiological situations, such as wound healing. VEGF-A plays a role in endothelial growth and angiogenesis, and bevacizumab is a humanized anti-VEGF-A mAb currently being studied for GC [10].

Cancer cells, through various immune checkpoint pathways, can escape the immune response. Release of tumour antigens can activate the innate and adaptive immune system, with several clone variants being developed through this process, which are then able to escape the immune-checkpoint system. Immune-checkpoint inhibitors can re-activate the immunogenic effect against cancer cells; programmed cell death protein-1 (PD-1) binds to the PD-L1/2 ligand and can stimulate apoptosis in T-lymphocytes, while CTLA-4 binds to CD28, producing further inhibitory signals. Pembrolizumab is an IgG4 humanized antibody which targets the PD-1 receptor of lymphocytes [11].

In the studies below, we will review the most up-to-date trials demonstrating the promising outcomes of the use of mAbs in gastro-esophageal cancer patients, as targeted-to-receptor treatments and/or immunoactivators (Figure 1).

## 2. Gastro-Esophageal Cancer Clinical Studies

### 2.1. The PD-1/PD-L1 Axis

The inhibition of the PD-1/PD-L1 pathway has shown significant results and durable responses, improving survival in these patients. In particular, Checkmate 648 was an open-label, phase III trial, which recruited a total of 970 advanced esophageal squamous cell carcinoma (ESCC) patients. The patients were randomized to receive either nivolumab, an anti-PD-1 mAb, plus chemotherapy (fluorouracil and cisplatin), or nivolumab plus ipilimumab, a CTLA4 mAb, or chemotherapy alone. After a 13-month minimum follow-up period, there was a superior overall survival (OS) observed with nivolumab plus chemotherapy, compared to chemotherapy alone, both among patients with tumour cell PD-L1 expression of 1% or greater (median, 15.4 vs. 9.1 months; hazard ratio (HR), 0.54; 99.5% confidence interval [CI], 0.37 to 0.80; *p* < 0.001) and in the overall population (median, 13.2 vs. 10.7 months; HR, 0.74; 99.1% CI, 0.58 to 0.96; *p* = 0.002). OS, which was the primary endpoint of the study, was also significantly longer with nivolumab plus ipilimumab than with chemotherapy, among patients with tumour cell PD-L1 expression of 1% or greater (median, 13.7 vs. 9.1 months; HR, 0.64; 98.6% CI, 0.46 to 0.90; *p* = 0.001) and in the overall population (median, 12.7 vs. 10.7 months; HR, 0.78; 98.2% CI, 0.62 to 0.98; *p* = 0.01). Favorable progression free survival (PFS) rates and a higher proportion of patients with prolonged response were observed with nivolumab plus chemotherapy and nivolumab/ipilimumab compared to chemotherapy alone, regardless of PD-L1 expression. Responses were seen in all three arms, regardless of PD-L1 expression level. No new safety signals were observed with either nivolumab plus chemotherapy or nivolumab/ipilimumab. The incidence of treatment-related adverse events (AEs) of grade 3 or 4 was 47% with nivolumab plus chemotherapy, 32% with nivolumab/ipilimumab, and 36% with chemotherapy alone. These data provide support for both immunotherapy-based regimens as first-line standard of care (SoC) in ESCC [12].

The efficacy of pembrolizumab, PD-1 inhibitor mAb, in the first-line setting, for patients with advanced GC, was the subject of Keynote 181. This trial was a randomized, double-blind, placebo-controlled, phase III study, which investigated the role of pembrolizumab, in combination with trastuzumab, an anti-HER2 mAb, in GC or gastro-esophageal junction adenocarcinomas, with HER2 amplification or overexpression, which occur in around 20% of advanced cancers. Patients received either SoC chemotherapy, FOLFOX (5-FU with oxaliplatin) or CAPOX (capecitabine with oxalipatin), together with trastuzumab and pembrolizumab, or SoC chemotherapy treatment, with trastuzumab and a placebo. Patients in the pembrolizumab arm had significantly prolonged OS and the best response rate, with 74% of patients achieving complete and partial response vs. 52% of patients in the placebo arm. Therefore, adding pembrolizumab to trastuzumab and chemotherapy, markedly reduced tumour size, induced complete response in some participants, and significantly improved the objective response rate (ORR) (74.4% in the pembrolizumab arm, compared to 51.9% in the placebo arm) [13].

In the randomized, phase III, Keynote 590 trial, the role of pembrolizumab was investigated, in the locally advanced, unresectable or metastatic EC (adenocarcinomas or squamous cell carcinomas), as well as in the advanced or metastatic esophagogastric cancer setting. First-line pembrolizumab plus chemotherapy (5-FU or cisplatin based), compared to chemotherapy alone, exhibited a sustained benefit, with 12 additional months of follow-up in locally advanced or metastatic EC patients. Benefits were also found for patients with a diagnosis of esophagogastric junction adenocarcinoma, with an ORR of 45%, compared to 29.3% for chemotherapy alone, after 12 additional months of follow-up. The treatment group had a median Duration of Response (DoR) of 8.3 months, compared to 6.0 months for chemotherapy alone. The median OS on the pembrolizumab arm was 12.4 months vs. 9.8 months on the chemotherapy arm, for the overall population, and 13.5 months for the PD-L1 CPS score ≥ 10 group, compared to 9.4 months for the chemotherapy arm. The investigators concluded that these findings provide additional evidence supporting the use of pembrolizumab plus chemotherapy as a new SoC first-line treatment option in advanced EC [14].

Tislelizumab, a humanized immunoglobulin G4 (IgG4) variant mAb against PD-1 has been used in combination with the anti-VEGF recombinant humanized mAb bevacizumab, plus CAPOX, in this ongoing, phase II, prospective, single-arm study. The addition of tislelizumab has demonstrated promising efficacy in PD-L1 < 5, HER2(−), locally advanced or metastatic gastro-esophageal cancer patients, with a manageable safety profile [15]. Similarly, in the RATIONALE-305 study, tislelizumab, together with chemotherapy, as first-line treatment for GC/gastro-esophageal junction cancer (GEJC), has continued to demonstrate clinically meaningful improvements in OS, PFS, and DoR compared to placebo plus chemotherapy, with no new safety signals, following a minimum 3-year follow-up period [16].

RATIONALE-306, investigated the role of tislelizumab in unresectable, locally advanced or metastatic ESCC. Patients were randomized to receive either tislelizumab, an anti-PD-1 mAb, with high affinity and specificity for PD-1, or SoC chemotherapy (cisplatin or oxaliplatin, plus fluoropyrimidine, or cisplatin or oxaliplatin with paclitaxel). Tislelizumab plus chemotherapy, as first-line treatment, demonstrated a statistically significant and clinically meaningful improvement in OS, compared to chemotherapy alone, in patients with advanced or metastatic ESCC. The median OS was 17.2 months in the tislelizumab arm, compared to 10.6 months in the SoC chemotherapy arm, with a HR of 0.66 (95% CI 0.54, 0.80; *p* < 0.0001) in all randomized patients. In patients with PD-L1 CPS score ≥ 10, the median OS was 16.6 months in the tislelizumab arm, compared to 10 months in the chemotherapy arm (HR 0.62; 95% CI 0.44, 0.86; *p* = 0.0020). There was a consistent OS benefit across all prespecified subgroups, including geographic regions, races, investigator-chosen chemotherapy options and PD-L1 expression status. The OS benefit with tislelizumab plus chemotherapy was accompanied by significant improvements in PFS and ORR, with a more durable tumour response, compared to placebo plus chemotherapy. Tislelizumab plus chemotherapy had a manageable safety profile in patients with advanced or metastatic ESCC, with no new safety concerns identified. Results of the RATIONALE-306 study support tislelizumab plus chemotherapy as a standard first-line therapy option for patients with advanced or metastatic ESCC [17,18].

The MOONLIGHT study investigated the combination of FOLFOX chemotherapy plus nivolumab and ipilimumab, administered in parallel, and it was found to be clearly more effective than FOLFOX induction chemotherapy followed by nivolumab and ipilimumab combination immunotherapy. Although associated with lower toxicity, the study did not support the use of sequential treatment in the first-line setting [19]. A summary of the studies can be found in Table 1.

### 2.2. The Case of HER2 Positive GC/EC

In the case of HER2 positive gastro-esophageal cancers, the role of trastuzumab and the recombinant trastuzumab deruxtecan, with the attached topoisomerase inhibitor, seem to have made a breakthrough in these types of cancer and their treatment approach.

The KEYNOTE 811 study investigated the role of pembrolizumab, in addition to trastuzumab and chemotherapy, in patients with HER2-positive EC or GEJC. Patients were enrolled into the study irrespective of their PD-L1 status; however, 88% and 85% of the patients in the pembrolizumab or placebo arms, respectively, had a PD-L1 combined positive score ≥ 1 at the interim timepoint. HER2-positive status is defined as immunohistochemistry (IHC) 3+ or IHC 2+ and in situ hybridization positivity. The patients were receiving CAPOX as chemotherapy agents, although the study also permitted 5-fluorouracil and cisplatin. The ORR for the pembrolizumab arm was 74.4% compared to 51.9% for the placebo arm, with a statistically significant difference of 22.7% (95% CI [11.2, 33.7]; *p* = 0.00006). Pembrolizumab also led to deeper responses, with 11% of patients in this arm achieving a complete response compared to 3% in the placebo arm. The most common treatment-related AEs reported were diarrhoea, nausea, and anaemia, with similar rates between the two groups. The immune-mediated AEs documented, notably pneumonitis and colitis, were more common in the pembrolizumab arm [20]. The final overall analysis showed that first-line pembrolizumab plus trastuzumab and chemotherapy provided a statistically significant and clinically meaningful improvement in OS vs pembrolizumab plus trastuzumab and chemotherapy, in all patients with unresectable, HER2+ metastatic GC/GEJC. OS was longer in patients with PD-L1 CPS ≥ 1. These data support the approval of pembrolizumab plus trastuzumab and chemotherapy in patients with HER2 positive GC/GEJC and confirm this regimen as SoC in the first-line setting [21].

Another interesting study was DESTINY-GASTRIC01. This trial verified the role of trastuzumab deruxtecan, the novel anti-HER2-targeted ADC, cleaved with the topoisomerase I inhibitor, compared to SoC chemotherapy treatment, in patients with HER2 positive GC/GEJC. A total of 51% of the patients in the trastuzumab deruxtecan group had an objective response, as compared to 14% in the physician’s choice group (*p* < 0.001). OS was longer with trastuzumab deruxtecan than with chemotherapy alone (median, 12.5 vs. 8.4 months; HR for death, 0.59; 95% CI, 0.39 to 0.88; *p* = 0.01). The most common AEs were neutropenia (in 51% of the trastuzumab deruxtecan group and 24% of the physician’s choice group), decreased total white-cell count (21% and 11%), and anaemia (38% and 23%, respectively). A total of 12 patients had trastuzumab deruxtecan-related interstitial lung disease (ILD) or pneumonitis (grade 1 or 2 in nine patients and grade 3 or 4 in three patients), with one documented drug-related death due to pneumonia, which was noted in the trastuzumab deruxtecan arm. However, it was concluded that therapy with trastuzumab deruxtecan led to significant improvements in response and OS, compared to SoC therapies, among patients with HER2 positive GC [22]. This was further followed by the DESTINY-Gastric 003 trial, which enrolled patients with HER2 positive (immunohistochemistry [IHC] 3+ or IHC 2+/in situ hybridization-positive by local testing) EC/GEJC/GC patients, globally, irrespective of PD-L1 status. Patients were randomized to SoC (fluoropyrimidine-based regimen), trastuzumab deruxtecan monotherapy or trastuzumab deruxtecan—based combinations and were stratified by HER2 status. The primary endpoint was ORR, as per the RECIST 1.1 criteria. It was found that the combination of trastuzumab deruxtecan with SoC chemotherapy and pembrolizumab had promising anti-tumour activity, warranting further study in these types of cancer. Tolerability was lower with the triplet combination; however, the overall safety profile was manageable [23].

Further interesting studies have demonstrated novel anti-cancer treatments for GC patients, with the use of the HER3-DXd in the global phase II study HERTHENA-PanTumor01 [24]. Another phase II trial investigated the role of ZV0203, a novel pertuzumab-based antibody–drug conjugate (ADC), in patients with HER2 positive advanced solid tumours, including GC, with positive preliminary results [25]. Further ADCs against HER2 have been investigated, including the newest drug IBI354 in this phase I study, with positive preliminary results [26].

The anti-HER2 ADC GQ1005 has shown some positive preliminary data in a phase Ia/Ib study, recruiting patients with HER2 expression (IHC2+/3+) GC (26 patients) among other HER2 cancer types (breast and lung). So far, GQ1005 demonstrates an excellent safety profile and encouraging anti-tumour activity in HER2-expressing tumours, including heavily treated HER2-positive GC, and further survivorship data are awaited [27].

The MAHOGANY study investigated the role of margetuximab, an Fc-optimized mAb that binds HER2, in combination with retifanlimab, a humanized IgG4 mAb, which binds to PD-1 and blocks its interaction with PD-L1/2, with or without chemotherapy, or in combination with tebotelimab, an IgG4κ bispecific DART^®^ molecule, which binds PD-1 and lymphocyte activation gene 3 concomitantly, disrupting these non-redundant inhibitory pathways, in order to further restore exhausted T-cell function. Targeting simultaneously HER2 and PD-1 (margetuximab plus retifanlimab) or HER2 and PD-1 plus LAG-3 (margetuximab plus tebotelimab) enhances anti-tumour activity and the innate or adaptive immune response, with promising potential for patients with unresectable or metastatic GC/GEJC [28].

In an ongoing phase II trial, zanidatamab, a HER2-targeted bispecific antibody is being tested, together with standard of care chemotherapy (5FU-Oxaliplatin-based chemotherapy. The trial is demonstrating a manageable safety profile and positive preliminary data, including median OS data, at 3.5 years follow-up of patients. Further primary and secondary endpoints are being investigated, and a further study, the global phase III HERIZON-GEA-01 trial, is investigating the combination of zanidatamab with chemotherapy and/or tislelizumab, as a first-line treatment of HER2-positive metastatic gastro-esophageal adenocarcinomas [29]. A summary of the studies can be found in Table 1.

### 2.3. The Case of FGFR2b

The FIGHT study investigated the role of bemarituzumab, combined with SoC chemotherapy (FOLFOX6) in advanced GC/GEJC. Bemarituzumab is a first-in-class humanized IgG1 mAb, selective for fibroblast growth factor receptor 2b (FGFR2b). Patients with FGFR2 amplification by ctDNA or FGFR2b amplification on IHC, with unresectable, locally advanced or metastatic GC were recruited to the study. Adding bemarituzumab to mFOLFOX6 was found to improve the OS of first-line FGFR2b+ GC patents, with a median OS of 19.2 months (95% CI: 13.6, not reached) compared to mFOLFOX6 alone, which was 13.5 months (95% CI: 9.3, 15.9). The outcomes favored bemarituzumab across the pre-specified subgroups. For the subset of patients with ≥10% FGFR2b+ by IHC, the median OS for bemarituzumab was 25.4 months (95% CI: 13.8, not reached) vs. 11.1 months (95% CI: 8.4, 13.8) for the placebo (HR: 0.41 95% CI: 0.23, 0.74). Patients with overexpression of FGFR2b, even without ctDNA amplification, demonstrated a benefit from the addition of bemarituzumab to mFOLFOX6, supporting further evaluation of bemarituzumab in tumours with FGFR2b overexpression, without the requirement for gene amplification [30]. A summary of the studies can be found in Table 1.

### 2.4. The Case of Claudin18.2

The FAST study investigated the role of zolbetuximab, a chimeric mAb which mediates specific killing of Claudin18.2 (CLDN18.2)-positive cells through immune effector mechanisms. CLDN18.2 is a specific marker only expressed on cancer cells and luminal gastric epithelial cells, making it a promising target for cancer therapy. Patients who were enrolled into the study were diagnosed with advanced GC/GEJC/EC (aged ≥18 years) and had moderate-to-strong CLDN18.2 expression in ≥40% tumour cells. Eligible patients received either epirubicin–oxaliplatin–capecitabine (EOX) treatment every 3 weeks, or zolbetuximab plus EOX every 3 weeks, with the primary endpoint being PFS and secondary endpoint OS. Adding zolbetuximab to first-line EOX provided longer PFS and OS vs. EOX alone, in both the overall population (PFS: HR = 0.44; 95% CI, 0.29–0.67; *p* < 0.0005; OS: (HR = 0.55; 95% CI, 0.39–0.77; *p* < 0.0005) and the subgroup of patients with moderate-to-strong CLDN18.2 expression (PFS: HR = 0.38; 95% CI, 0.23–0.62; *p* < 0.0005) [31].

In another first-in-human phase I/II trial, the ADC CLDN18.2 (RC118) was tried in 18 GC/GEJC patients, with preliminary data verifying an overall manageable safety profile. Ongoing investigation of RC118 is testing its combination with toripalimab, while establishing the optimal dose regimen [32].

Another bispecific antibody targeting CLDN18.2 and CD47, in combination with chemotherapy and/or pembrolizumab, was investigated in the phase I/II study TWINPEAK, recruiting GC/GEJC patients. Patients with more than 10% CLDN18.2-positive tumour cells were eligible. Normally, it is thought that CLDN18.2 is expressed in approximately 70% of GC/GEJC patients, showing that it remains a negative prognostic factor for these patients. PT886 is an IgG1 bispecific antibody, targeting CLDN18.2, that mediates antibody-dependent cellular cytotoxicity, and overall immune-activation, facilitating tumour anti-cancer therapy [33].

Further interesting studies have demonstrated novel anti-cancer treatments for GC patients, including the use of the novel CLDN18.2 ADC XNW27011, used in patients with metastatic solid tumours, including GC patients [34]. In the phase I study SHR-A1904, a novel ADC comprising an IgG1 mAb-targeting CLDN18.2 was assessed in 73 GC/GEJC patients. Preliminary data show that this novel ADC has positive anti-tumour activity, with ORR and DCR values of 55.6% and 88.9%, respectively, for the 6.0 mg/kg dose, and 36.7% and 86.7%, respectively, for the 8.0 mg/kg dose, with a manageable safety profile. Grade 3 or higher treatment-related AEs occurred in 53.4% of patients, with the most common (>10%) AEs being decreased WBC count, neutropenia and anaemia. Overall, the authors indicate some very promising initial data and further outcomes are awaited [35].

Another interesting study has been initiated and investigates the use of FG-M108, an afucosylated IgG1 mAb targeting CLDN18.2 in combination with CAPOX, as the first-line treatment. The preliminary data have been announced and show a manageable safety and toxicity profile, with some improvement in the overall efficacy outcomes. Further results are awaited from this study [36]. A summary of the studies can be found in Table 1.

### 2.5. Other Promising Agents

Further early-stage studies have been investigating other novel anti-cancer agents, with promising preliminary results, for the treatment of EC and GC in the advanced setting, compared to SoC treatments. Fruquintinib is an inhibitor of VEGF receptor-1/2/3, which was administered together with paclitaxel, in the multicenter, double-blind phase III FRUTIGA trial, which recruited 703 patients with advanced GC/GEJC, after progression to fluorouracil- and platinum-containing chemotherapy. Patients were randomized in a 1:1 ratio to receive fruquintinib or placebo plus paclitaxel, and the study met its primary endpoint of significant median PFS, which was 5.6 months in the fruquintinib arm compared to 2.7 months in the placebo arm (HR 0.57; 95% CI, 0.48–0.68; *p* < 0.0001). The median OS was better in the fruiquintinib arm (9.6 months vs. 8.4 months) but was not statistically significant (HR 0.96, 95% CI, 0.81–1.13; *p* = 0.6064). The treatment was well tolerated, although there were some grade 3 or higher AEs reported, including haematological toxicities. The combination of fruquintinib plus paclitaxel, as a second-line treatment in patients with advanced GC/GEJC, shows some benefit from this alternative combination treatment option [37]. A further post-hoc analysis was conducted to evaluate the efficacy of the fruquintinib/paclitaxel combination in patients who received prior immunotherapy within the FRUTIGA study. From a total of 703 patients enrolled in the study, 82 had received prior immunotherapy treatment, of whom 35 were treated with fruquintinib/paclitaxel. The most commonly used prior immunotherapy treatments were sintilimab and tislelizumab. Overall, it was shown that patients who received prior immunotherapy treatment had a further reduction of progression or death rates, compared to those who received prior chemotherapy only, and that the combination of fruquintinib/paclitaxel in these immunotherapy pre-treated patients had further clinical value in overall efficacy [38].

Another interesting phase I study is using, for the first time in humans, the molecule CTS2190, a type I protein arginine methyltransferases (PRMTs) inhibitor. The preliminary results have demonstrated significantly reduced intra-tumour asymmetric dimethylarginine (ADMA) activity levels and limited oncogenic proliferation, by epigenetic modulation, RNA splicing and DNA damage, as well as remarkable cancer immune activation in various solid and/or hematological cancers. The dose escalation results from this ongoing first-in-human phase I/II study have been revealed, and they evaluate the safety, tolerability, pharmacokinetics (PK) and efficacy of CTS2190, in patients with advanced solid tumours, while further results are awaited [39].

The Gemini study (an open-label, multicenter, phase II study) is investigating the role of rilvegostomig, a bispecific humanized IgG1 mAb against PD-1 and TIGIT, in combination with SoC chemotherapy, CAPOX (oxaliplatin + capecitabine) or FOLFOX (oxaliplatin + 5-FU + folinic acid), in patients with locally advanced, unresectable or metastatic GC. This is an ongoing study; however, the preliminary data are very reassuring, verifying evidence of efficacy and a manageable toxicity profile, for the 40 patients enrolled into the study, with 27 of them continuing treatment at data cut-off, with a confirmed ORR of 52.5%. No grade 5 toxicities were reported in the study [40].

Another study investigated the role of SHR-1701, a bifunctional agent composed of an IgG4 mAb, targeting PD-L1, fused with the extracellular domain of TGF-βIIR, in addition to SoC chemotherapy, for unresectable, locally advanced or metastatic HER2-negative GC/GEJC patients. Median follow-up was 8.5 months at data cut-off, with a median OS significantly prolonged with the addition of SHR-1701 to chemotherapy, vs placebo plus chemotherapy, in patients with PD-L1 CPS ≥ 5. Further survivorship data and assessment of ongoing safety and toxicity are awaited, but so far, the results are very promising [41].

Several new studies are in set-up, and many promising survivorship data and overall outcomes are awaited, including those for the phase Ib/II FUNCTION study. In this multicenter trial, fruquintinib, a highly selective oral tyrosine kinase inhibitor of vascular endothelial growth factor receptors (VEGFRs) 1, 2, and 3, is used in combination with sintilimab, an NMPA-approved PD-1 inhibitor, and SoC chemotherapy (CAPOX). Upon completion of chemotherapy cycles, fruquintinib plus sintilimab will be administered as maintenance treatment. Further survivorship data and overall response rates are awaited [42]. The TROP2 ADC sacituzumab tirumotecan (sac-TMT) is being investigated in another phase III study, with some preliminary data demonstrating ongoing anti-cancer activity, for patients with previously treated metastatic GEJC, compared to treatment of physician’s choice [43].

Another interesting phase I study is investigating for the first time in humans the use of DS-9606a, an ADC composed of a humanized anti-CLDN6 antibody. CLDN6 is involved in cell-to-cell tight junctions and this ADC, which involves a cleavable linker and pyrolobenzodiazepine payload, has demonstrated a favorable safety profile, and further results are awaited as the study is ongoing [44]. Another prospective, open-label, multi-center, adaptive phase II trial, PLATFORM, has assessed maintenance therapy in patients with advanced esophagogastric adenocarcinoma after platinum-based first-line induction chemotherapy. Patients were randomized 1:1 to surveillance or maintenance capecitabine, with the addition of ramucirumab, a VEGFR receptor 2 mAb, with a median OS of 19.5 months in the capecitabine–ramucirumab arm. This demonstrates an OS benefit for patients having previously received first-line chemo-immunotherapy treatment, offering a potential new therapeutic option for these patients [45]. Another phase I trial checked the safety and tolerability of the ADC ABBV-400, consisting of the c-Met-targeting mAb telisotuzumab, conjugated to a novel topoisomerase 1 inhibitor payload. Patients with a background of advanced gastro-esophageal adenocarcinoma who have progressed after ≤2 prior cytotoxic chemotherapeutic regimens, received ABBV-400 every three weeks. The tolerability, dose escalation and safety profile were initially investigated. ABBV-400 has demonstrated a tolerable safety profile and anti-tumour activity in patients with advanced gastro-esophageal adenocarcinoma, while ongoing studies are being carried out to further investigate its efficacy in patients with GEA [46]. A summary of the studies including these new promising agents can be found in Table 2.

## 3. Conclusions

There is a new era of targeted agents against particular genes and protein-signalling pathways that have changed the picture in the treatment of gastro-esophageal cancers. These agents are paving the way towards a more personalised approach in oncological management of patients with gastro-esophageal cancers. The role of tumour microenvironment and cancer molecular genetics can be crucial in the way we understand and more efficiently apply the use of mAbs, towards further optimization of treatment strategies, which could lead to further therapeutic advancements in the field.

Lately, approaches have emerged that combine direct tumour cell cytotoxicity with immunological activation and/or removal of tumour-elicited immunosuppression, but much research is still needed to elucidate the mechanisms behind anti-tumour effects of particular treatment combinations compared to others. Preliminary OS data are a useful measure of efficacy, but many more data are required, in terms of PFS, ORR and long-term survivorship results, together with quality of life and toxicity profile results. The design of future studies might elaborate and address further needs and the rationale behind the use of particular combination treatments, while providing more information regarding long-term efficacy and overall safety issues raised.

The role of diagnostic biomarkers, such as MSI, EBV, PD-L1 and HER2, and their expression levels, can be pivotal in the diagnosis and better therapeutic management of those types of cancers. Immune checkpoint inhibitors, in combination with SoC chemotherapy, and targeted treatments, seem to be paving the way towards a new era in the treatment of gastro-esophageal cancers.

## Figures and Tables

**Figure 1 ijms-26-01090-f001:**
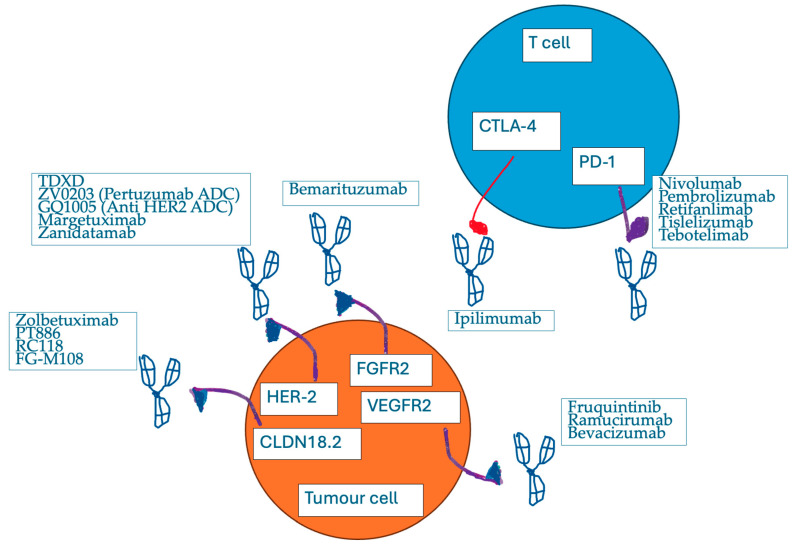
The interaction between tumour cell receptors and the bi-specific mAbs, targeting various receptors or acting as immunoactivators.

**Table 1 ijms-26-01090-t001:** Summary of trials using molecules targeting the PD-1/PDL-1 axis, HER2, FGFR2, CLDN18.2.

Trials	Agents	Outcomes	Toxicity
**Checkmate648**	Ipilimumab/Nivolumab (*PD-1/PDL-1 axis*)	Median OS 15.4 months: nivolumab plus chemo vs. 9.1 months chemo alone, both among patients with tumour cell PD-L1 expression of 1% or greater HR, 0.54; 99.5% [CI], 0.37 to 0.80; *p* < 0.001)	Manageable safety and toxicity profile (low rate of G3 or higher toxicities on all trials)
**Keynote 181**	Pembrolizumab (*PD-1/PDL-1 axis*)	ORR 74.4% in the pembrolizumab arm vs. 51.9% in the placebo arm	
**Keynote 590**	Pembrolizumab (*PD-1/PDL-1 axis*)	Median OS on the pembrolizumab arm: 12.4 months vs. 9.8 months on the chemotherapy arm	
**RATIONALE-306**	Tislelizumab (*PD-1/PDL-1 axis*)	Median OS 17.2 months in the tislelizumab arm, compared to 10.6 months in the SoC chemotherapy arm: HR of 0.66 (95% CI 0.54, 0.80); *p* < 0.0001	
**MOONLIGHT**	Ipilimumab/Nivolumab (*PD-1/PDL-1 axis*)	Clinically significant improvement in OS and PFS with addition of PD-1/PDL-1 inhibition compared to chemotherapy alone	
**KEYNOTE 811**	Trastuzumab/Pembrolizumab (*PD-1/PDL-1 axis*)	ORR for the pembrolizumab arm: 4.4% vs. 51.9% for the placebo arm (95% CI [11.2, 33.7]; *p* = 0.00006)	
**DESTINY-GASTRIC 01/03**	Trastuzumab Deruxtecan (*HER2*)	Median OS TDxD: 12.5 months vs. 8.4 months chemo alone; HR: 0.59; 95% CI, 0.39 to 0.88; *p* = 0.01)	
**HERIZON-GEA-01**	Zanidatamab/tislelizumab (anti-HER2)	Clinically significant improvement in OS and PFS with addition of PD-1/PDL-1 inhibition compared to chemotherapy alone	
**FIGHT**	Bemarituzumab (anti-FGFR2b)	Median OS of 19.2 months (95%CI: 13.6, not reached) for Bemarituzumab plus mFOLFOX6, compared to 13.5 months for mFOLFOX6 alone (95%CI: 9.3, 15.9).	
**FAST**	Zolbetuximab (anti-CLDN18.2)	Zolbetuximab plus EOX provided longer PFS and OS versus EOX alone, in both the overall population (PFS: [hazard ratio (HR) = 0.44; 95% confidence interval (CI), 0.29–0.67; *p* < 0.0005], (OS: (HR = 0.55; 95% CI, 0.39–0.77; *p* < 0.0005)) and the subgroup of patients with moderate-to-strong CLDN18.2 expression (PFS: (HR = 0.38; 95% CI, 0.23–0.62; *p* < 0.0005).	
**OTHERS:** **RATIONALE-305** **FIH** **HERTHENA-PanTumor01** **MAHOGANY** **RC118** **TWINPEAK**	Tislelizumab/Bevacizumab anti-HER2 ADC GQ1005 ZV0203 Pertuzumab ADC Margetuximab/Retifanlimab or Margetuximab/ Tebotelimab CLDN18.2 ADC/Torpalimab Pembrolizumab/ PT886 IgG1 bispecific antibody targeting CLDN18.2	Preliminary data confirm clinically meaningful improvement in OS and PFS with trial agent vs. SOC	

**Table 2 ijms-26-01090-t002:** Summary of new promising studies investigated.

Other New Promising Studies	Agents	Preliminary Data/Outcomes
**FRUTIGA**	Fruquintinib (VEGFR inhibitor)	Median PFS 5.6 months in the fruquintinib arm vs. 2.7 months in the placebo arm (HR 0.57; 95%, CI 0.48–0.68; *p <* 0.0001).
**ABBV-400**	Telisotuzumab (conjugated to a novel topoisomerase 1 inhibitor payload)	Tolerable safety profile (low percentage of G3 or higher adverse events). Preliminary efficacy results show promising anti-tumour activity, further outcomes awaited.
**GEMINI**	Rilvegostomig	
**SHR-1701**	Bifunctional agent composed of an IgG4 mAb, targeting PD-L1	
**FUNCTION**	Fruquintinib-Sintilimab	
**PLATFORM**	Ramucirumab	
**CTS2190**	PRMT inhibitor	
**TROP2 ADC**	Sacituzumab tirumotecan (sac-TMT)	
**DS-9606a**	DS-9606a (antiCLDN6 Ab)	

## Data Availability

All data are publically available and sited accordingly. The figures and table provided in the manuscript are original.

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
