# Peer review of "Monoclonal Antibodies in Metastatic Gastro-Esophageal Cancers: An Overview of the Latest Therapeutic Advances"

_ijms, 2025, doi:10.3390/ijms26031090_

Round 1

Reviewer 1 Report

Comments and Suggestions for Authors

In this review, new approaches to the treatment of esophageal and gastric cancers using monoclonal antibodies and ADCs are describe in detail to direct readers to their use in clinical trials.  Their major conclusion is that when used in combination with chemotherapy, antibody based immunotherapy provides improved responses over chemotherapy treatments alone.  To provide a clear review, the authors organized the results of clinical trials for specific antigenic targets directed against the PD-1/PD-L1 axis, HER2, FGFR2b, and Claudin 18.2.  This reviewer has the following comments for the authors:

(1) The review is comprehensive with a good literature bibliography.  It clearly was written for oncologists who treat these types of tumors and by so doing, does not delve into the rationale for each trial design.  Hence, no discussion of the contribution of immunotherapy is attempted other than its role as a cytotoxic for tumor.  

(2) I like the organization around tumor antigen targets which is a good way of tabulating what trials are in progress (see Table 1).   This enables the authors to show which targets are actively being pursued for both antibodies and ADCs.  The discussion also clearly provides strong evidence to show that combinations of antibody treatments and chemotherapy are more effective.

(3) Missing, in my opinion, is why certain combinations are showing more promise.  For instance, approaches that combine direct tumor cell cytotoxicity with immunological activation and/or removal of tumor-elicited immunosuppression  via anti-Treg, anti-MDSC, reversal of T-cell exhaustion, etc. would give the reader a handle of why these combinations are being tried.  In addition, no information is provided as to why treatment failed.  OS data are useful as a measure of tumor treatment success but gives no information about what is needed to make each approach better.  This type of information may not be included in clinical trial results and, if so, the authors should discuss how we can redesign future trials to uncover this type of critical data.

(4) one typo: page 3, line 97: "This" should read "These" (pleural).

Author Response

Reply to reviewer 1:

We thank the reviewer for their time and valuable comments and we have enclosed the answers to the points raised, as seen below.

-Points 1-3 have been taken into consideration and addition of a paragraph explaining the rationale behind the studies and the need for similar trials has been added on discussion section:

“Lately, there have been approaches that combine direct tumor cell cytotoxicity with immunological activation and/or removal of tumor-elicited immunosuppression, but a lot more is still needed to elucidate the mechanisms behind anti-tumour effect with combination of particular treatments, rather than others. Preliminary OS data are a useful measure of efficacy, but a lot more data are required, in terms of PFS, ORR and long-term survivorship results, together with quality of life and toxicity profile results. Design of future studies might elucidate further needs and the rationale behind the use of particular combination treatments and it will provide with more information regarding long term efficacy and safety data”.

 This is a review describing the latest clinical trials in gastro-oesophageal cancers, which indeed is in the field of clinical research oncology. The need for these trials is related to the poor prognosis of these types of cancers and the new emerging knowledge of molecular targets and the new advances in pharmacology, with the introduction of targeted treatments and the evolution of the monoclonal antibodies. Therefore, the standard of care chemotherapy options will be in the future enhanced with or replaced by precision medicine approaches (targeted anticancer treatments/immunoactivators and monoclonal antibodies). This article has been an effort to depict the existing trials, most of them having only preliminary data available as they are at a very early phase but shows the potential of monoclonal antibodies for the treatment of GOC.

-Point 4 has been addressed on text (This replaced by these).

Reviewer 2 Report

Comments and Suggestions for Authors

This review describes clinical trials employing monoclonal antibodies and their derivatives for the treatment of metastatic gastro-esophageal cancers.

Although the text provides an exhaustive description of the status of the art for the treatment of these cancers through monoclonal antibodies, the review would benefit from figures and additional information in the tables.

I suggest to author to include the following in the revised manuscript:

1) Include a figure 1 depicting tumor targets in gastro-esophageal cancers described in the text and in table 1, including their role in tumor development, progression, survival, metastasis etc and which mAb alone or in combination is used to block each target 

2) Information included in table 1 and 2 need to be more detailed. I suggest to include specific information for each clinical trial. Example: checkmate 648. Add a column describing which cancer is targeting each clinical trial; include more details in outcomes for each clinical trial (in addition to clinical significant improvement in OS or PFS, please add how many months, compared to what etc ); add a column including more details in safety and toxicity profile for each clinical trial

3) Include a figure 2 after table 1,  depicting tumor targets in gastro-esophageal cancers described in the text and in table 2, including their role in tumor development, progression, survival, metastasis etc and which mAb alone or in combination is used to block each target

Author Response

Reply to reviewer 2:

We thank the reviewer for their time and valuable comments and we have enclosed the answers to the points raised, as seen below.

-A figure has been included, portraying the tumour cell with its receptors and the T cell with the immunoreceptors (CTLA-4, PDL1), depicting the immune-activation and the targeting of receptors with monoclonal antibodies (including HER2, VEGFR, Claudin 18.2, FGFR2, as mentioned on the paper).

-Further information, based on publicly available data has been added on the table. However, a lot of the trials are still of a very early phase and only preliminary data is available, therefore we do not have all these details publicly available yet, hence we cannot include that on the table.

-The figure added already after table 1 shows the immunoactivation and targeting of receptors of the cancer cells as explained on the trials on the text and hopefully that will cover sufficiently the reader of the paper.

Round 2

Reviewer 2 Report

Comments and Suggestions for Authors

I thanks the author for the answers to my questions.

Just a couple of things:

1) I suggest authors to organize the table 1 in 3 columns for a better comprehension. First column: just the name of the trial; second column:  outcome of each trial (just move information you added in red in this column); third column: safety and or toxicity data

2) Figure 1 seems to be blurry. Please check the quality of the image. In addition, for each target authors should add the name of the antibody under the picture of each antibody.

Author Response

We thank the reviewer for their comments.

Table 1 has been adjusted with new columns added as instructed. The same was applied to table 2.

Figure 1 has been re-edited with captions.

Hopefully the desired outcome has been achieved, many thanks for your input.